# Mood disorders in children following neonatal hypoxic-ischemic encephalopathy

**María Álvarez-García**[1], **Isabel Cuellar-Flores**[2], **Purificación Sierra-García**[3‡], **José Martínez-Orgado**[2‡] *

**1** International PhD School, PhD Program in Health Psychology, UNED, Madrid, Spain, **2** Department of Neonatology, Hospital Clínico San Carlos-IdISSC, Madrid, Spain, **3** Department of Developmental and Educational Psychology, School of Psychology, UNED, Madrid, Spain

‡ PSG and JMO are contributed equally to this work as co-senior authors on this work.
* jose.martinezo@salud.madrid.org

## Abstract

### Background

Few studies on the consequences following newborn hypoxic-ischemic encephalopathy (NHIE) assess the risk of mood disorders (MD), although these are prevalent after ischemic brain injury in adults.

### Objective

To study the presence of MD in children survivors of NHIE.

### Methods

14 children survivors of NHIE treated with hypothermia and without cerebral palsy and 15 healthy children without perinatal complications were studied aged three to six years for developmental status (Ages and Stages Questionnaire 3 [ASQ-3]) and for socio-emotional status (Preschool Symptom Self-Report [PRESS] and Child Behavior Checklist [CBCL] 1.5–5 tests). Maternal depression was assessed using Montgomery-Asberg Depression Rating Scale (MADRS). Socio-economic factors such as parental educational level or monthly income were also studied.

### Results

NHIE children did not present delay but scored worse than healthy children for all ASQ3 items. NHIE children showed higher scores than healthy children for PRESS as well as for anxious/depressive symptoms and aggressive behavior items of CBCL. In addition, in three NHIE children the CBCL anxious/depressive symptoms item score exceeded the cutoff value for frank pathology ($P = 0.04$ vs healthy children). There were no differences in the other CBCL items as well as in maternal MADRS or parental educational level or monthly income. Neither ASQ3 scores nor MADRS score or socio-economic factors correlated with PRESS or CBCL scores.

**Citation:** Álvarez-García M, Cuellar-Flores I, Sierra-García P, Martínez-Orgado J (2022) Mood disorders in children following neonatal hypoxic-ischemic encephalopathy. PLoS ONE 17(1): e0263055. https://doi.org/10.1371/journal.pone.0263055

**Data Availability Statement:** Deidentified individual participant data (including data dictionaries) will be made available, in addition to study protocols, the statistical analysis plan, and

the informed consent form. Data will be made available upon publication to researchers who provide a methodologically sound proposal for use in achieving the approved proposal's goals. Proposals should be submitted to the corresponding author as well as to the Hospital Clínico San Carlos' Clinical Research Ethics Committee (ceic.hcsc@salud.madrid.org).

**Funding:** This project received funds from the program ECL.NEONAT from the Biomedical Research Foundation of the Hospital Clinico San Carlos. The funder had no role in study design, data collection and analysis, decision to publish, or preparation of the manuscript.

**Competing interests:** The authors have declared that no competing interests exist.

## Conclusions

In this exploratory study children survivors of NHIE showed increased risk of developing mood disturbances, in accordance with that reported for adults after brain ischemic insults. Considering the potential consequences, such a possibility warrants further research.

## Introduction

Neonatal hypoxic-ischemic encephalopathy (NHIE) affects 1.5-3/1000 live newborns [1]. Despite the high impact on emotional and psychological wellbeing in infancy, most studies on the consequences following NHIE focused on motor and cognitive domains as well as on behavioral problems, mainly attention deficits and hyperactivity [2]. However, studies about the development of mood disorders (MD) in children surviving NHIE are scarce. This is striking taking into account that MD are a frequent complication of acute ischemic brain injury (IBI) in adults, which affect close to a third of those patients [3, 4]. Such disorders appear the following five years after the ischemic injury, peak three to five months post-insult, and consist mainly of depression but also anxiety, dysthymia, and adjustment disorders [3, 4]. The importance of such complications is stressed by the fact that MD following acute IBI in adults, in particular depression, are associated with increased mortality, worse quality of life and impaired physical and cognitive recovery in the short and long term [4, 5].

Reasons for the paucity of studies on the socio-emotional consequences of NHIE are multiple. There is considerable resistance to acknowledging that children may suffer MD or emotional psychopathology [6]. Assessment of MD in pre-school children is particularly difficult due to the intrinsic characteristics of this period [7, 8] The most worrying consequence of NHIE, cerebral palsy (CP), is associated with an increased presence of MD as in any chronic disabling condition leading to a significant loss of quality of life for patients and their parents [9]. Those emotional consequences of disability because of CP can mask the effects of NHIE by itself on development of MD when NHIE survivors developing CP are included in the study.

Psychopathology in children is often organized around three broad domains: internalizing, externalizing, and dysregulation symptoms [10]. One study in three-year-old children survivors of mild-to-moderate NHIE without CP reported increased anxiety and aggressive behavior [11]. However, children in that study did not receive hypothermia as a treatment for NHIE, which makes those results of limited value for current babies receiving hypothermia for moderate-to-severe NHIE as a standard of care [1]. A more recent study [12] reported increased anxiety/depression symptoms as well as sleep disorders in two-year-old children with minor neurologic symptoms after NHIE treated with hypothermia. However, that study does not compare NHIE children with healthy children but rather NHIE children with minor neurologic symptoms versus NHIE children without a neurologic symptom study [12].

The aim of this work was to study the presence of MD in children admitted as newborns to be treated with hypothermia for NHIE. We then compared that population with healthy children with no history of problems as newborns, and assessed those contextual factors that could potentially influence the development of such disorders.

## Methods

The study, comparing the characteristics of neonatal HIE survivors with children of the same age without perinatal negative events, was approved by the Hospital's Clinical Research Ethics

Committee (December 2018, record 12.1/18, CI: 18/534-E). Families were invited to participate in the study. They were informed of the study's general objective and procedure. All parents who agreed to participate signed the informed consent form.

## Study population and sample selection

Children included in the study were infants born at more than 35 weeks of gestational age, admitted to the Hospital's Neonatal Intensive Care Unit between January 2013 and December 2017 for consideration of hypothermia treatment because of moderate-to-severe NHIE. Severity of NHIE was classified following the modified Sarnat and Sarnat staging [13]. Another inclusion criteria was that the baby should have been included in our multi-disciplinary follow-up program at discharge. Exclusion criteria were death before discharge, major birth defects, no inclusion in hypothermia treatment, development of cerebral palsy during follow-up or parents refusing follow-up and/or with significant Spanish language limitations. In all babies from the study group a MRI scan was performed five to eight days after the hypoxic-ischemic event. MRI scan was repeated in all babies by the age of two to three years. By the time of the study, parents of selected patients were contacted by phone call to schedule a special visit to perform the assessment interview.

Once study patients were selected, babies > 35 weeks of gestational age with no noticeable perinatal complication, born during the same period and matched for gestational age, birth weight, sex and date of birth were identified in the clinical records. Parents were contacted by phone call to be invited to voluntarily attend a special visit to perform the assessment interview.

## Assessment

At the age of three to six years, a special visit was scheduled, unique for the control group and in addition to the usual follow-up in the case of NHIE babies. During that visit, the assessment protocol was completed by children and by parents. The assessment interview was performed by two experienced clinical psychologists, who were not blinded to the experimental group.

**Demographics.** Educational level was scored as follows: 1 = no studies; 2 = primary education; 3 = secondary education; 4 = professional training; 5 = high school; 6: higher professional training; 7: university.

Income per month was scored as follows: 1 = no income; 2 = unemployment subsidy; 3 = €500–1000; 4 = €1000–1500; 5: €1500–2000; 6 = more than €2000.

**Development.** Parents answered the Ages and Stages Questionnaire 3 (ASQ-3) [14] to assess the child's development in several domains (communication, fine and gross motor skills, problem solving and personal-social skills). ASQ3 can be used in children aged one month to six years and have a similar predictive value to the Bayley III test [12].

**Emotional assessment.** The socio-emotional status of children was assessed using two different approaches. Parents completed the Child Behavior Checklist (CBCL) for children aged one and a half to five years from the Achenbach System of Empirically Based Assessment ASEBA [15] to assess difficulties in several sets of behaviors (emotionally reactive, anxious/depressed, somatic complaints, withdrawn, sleep problems, attention problems, aggressive behavior, other problems) grouped into two broad bands of behavioral problems (Internalizing and Externalizing Problems). Children completed the Preschool Symptom Self-Report (PRESS), a pictorial instrument that assesses, with self-informed responses from identification with images, the presence of depression symptoms in preschool children [16].

Parents also answered the Montgomery-Asberg Depression Rating Scale (MADRS) [17] to assess the presence of depression symptomatology and its intensity.

### Statistical analysis

Data distribution was assessed using the Shapiro-Wilks test and found to follow a non-parametric distribution. Therefore, data were compared using the Mann-Whitney-U for comparisons between two groups. Contingency tables were analyzed using $X^2$. Data were shown as mean (95% CI) and analyzed with a statistical software package (GraphPad Prism 5; GraphPad Software, San Diego, CA, USA). A value of $P<0.05$ was considered statistically significant.

## Results

Study population flowchart is shown in Fig 1. Eventually 14 NHIE and 15 healthy children were assessed age three to six years. Of the 14 NHIE children, 11 had a normal MRI scan five to eight days after birth, whereas three babies showed a small focal lesion in the temporal or parieto-occipital cortex (Fig 2). None of the newborns showed any damage in the Basal Ganglia, Hypothalamus, Hippocampus or frontal cortex. All NHIE children showed normal MRI scans age two to three years.

### Sociodemographic and clinical profile

Control and NHIE groups were similar in gestational age, birth weight, sex distribution and age at assessment (Table 1). One baby from the NHIE group and two babies from the control group were born at 36 weeks of gestational age; the other were born at 37 weeks or more.

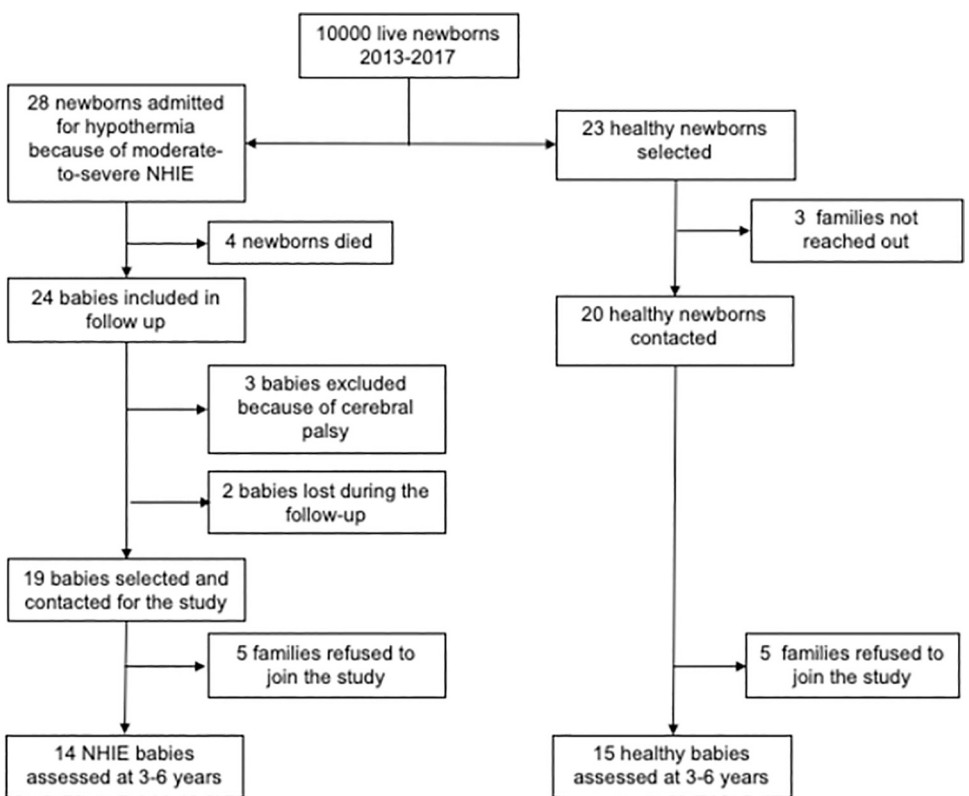

**Fig 1. Study population.** Babies eligible for the study were admitted to the neonatal Intensive Care Unit for consideration of hypothermia treatment for moderate-to-severe hypoxic-ischemic encephalopathy. Healthy babies with no perinatal problems matched for gestational age, birth weight, sex, and date of birth were selected for the control group.

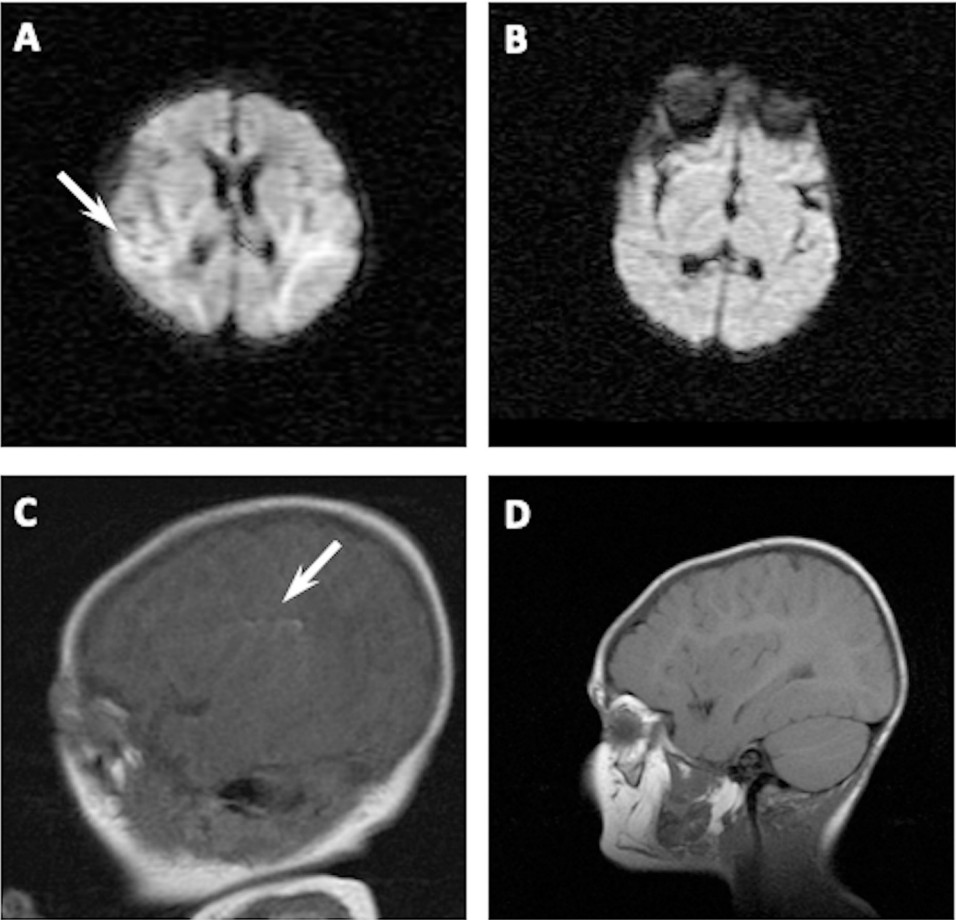

**Fig 2. Representative MRI scans showing altered studies in two hypoxic-ischemic newborns.** In A, the MRI scan performed at postnatal day 5 in a female (39 weeks of gestational age) showed increased signal in both temporal lobes using Diffusion-Weighed Images; scan performed in the same patient aged 30 months showed no abnormalities (B). In C, the MRI scan performed at postnatal day 5 in a male (41 weeks of gestational age) showed linear hyperintensity in left insular subcortical area using T1-Weighed Images; scan performed in the same patient aged 26 months showed no abnormalities (D).

**Table 1. Demographic data of control group and NHIE group children.**

|  | Control (n = 15) | NHIE (n = 14) | P value |
|---|---|---|---|
| Age at study (months) | 54.0 (47.5, 60.5) | 46.0 (42.9, 56.5) | 0.30 |
| Gestational age (weeks) | 39.0 (37.7, 39.9) | 39.0 (37.7, 39.4) | 0.48 |
| Birth weight (g) | 3210 (2817, 3558) | 3040 (2912, 3520) | 0.49 |
| Male/female | 11/4 | 10/4 | 0.91 |
| Educational level[a] |  |  |  |
| Mother | 7.0 (5.7, 7.0) | 7.0 (5.4, 7.1) | 0.82 |
| Father | 7.0 (6.1, 7.3) | 7.0 (5.3, 7.3) | 0.26 |
| Income per month[b] | 6 (5.5, 6.1) | 6 (4.9, 5.9) | 0.09 |

Median (95% CI). Statistical analysis using the Mann-Whitney test.

NHIE: newborn hypoxic-ischemic encephalopathy.

**Table 2. Results from testing with the ASQ3.**

|  | Control (n = 15) | NHIE (n = 14) | P value |
|---|---|---|---|
| Communication skills | 55 (51.1, 56.3) | 45 (42.9, 52.0) | 0.006 |
| Gross motor skills | 60 (57.8, 60.1) | 55 (50.9, 57.66) | 0.009 |
| Fine motor skills | 55 (53.6, 58.4) | 47.5 (40.0, 53.5) | 0.02 |
| Problem solving | 60 (57.4, 60.5) | 57.5 (50.8, 58.4) | 0.02 |
| Personal-social | 60 (51.9, 60.7) | 50 (44.9, 53.6) | 0.002 |

Median (95% CI). Statistical analysis using the Mann-Whitney test.

ASQ3: Ages and Stages Questionnaire-3.

NHIE: newborn hypoxic-ischemic encephalopathy.

Furthermore, socio-economic characteristics such as educational level or income per month were similar in both groups (Table 1). Children from both groups attended regular school at the time of the assessment interview. All children from the NHIE group were included by protocol in Early Intervention programs after discharge from the Intensive Care Unit.

## Development

Children from the NHIE group showed lower scores for all ASQ3 test items than children from the control group (Table 2). However, although performance was worse, mean values of all ASQ3 items in NHIE children did not attain the cutoff value to define a "delay", according to ASQ3 staging [14].

## Emotional assessment

The self-informed PRESS responses indicated that NHIE children had a greater presence of depression symptoms than children from the control group (PRESS score 0.00 [-0-18, 1.78] vs 2.00 [1.26, 4.42] points, median [95% CI] for the control and NHIE group, respectively, Mann-Whitney $P$ = 0.01).

The CBCL test did not reveal differences between NHIE and control children with the exception of two items: anxious/depressed symptoms, among internalizing problems, and aggressive behavior, among externalizing problems, in which NHIE children showed a higher score than control group children (Table 3). Moreover, after considering the accepted cutoff values for CBCL staging [15] three out of the 14 NHIE children versus none of the control group children obtained such a score in the anxious/depressed symptoms item as to be considered of clinical concern ($X^2$ = 3.88, $P$ = 0.04). In the case of aggressive behavior, two NHIE children and no control group children scored at values considered of clinical concern ($X^2$ = 2.30, $P$ = 0.12)

There were no differences in the score obtained in the depression scale (MADRS) between mothers from NHIE or control group children (MADRS score: 2.00 [1.42, 3.89] vs 4.00 [3.02, 10.20] points, median [95% CI] for control and NHIE, respectively, Mann-Whitney $P$ = 0.23).

## Factors associated with psychological symptoms in children diagnosed with HIE

After considering the items showing a poorer score in NHIE than in the control group children, PRESS and anxious/depressed symptoms and aggressive behavior in CBCL, no correlation was found between those items and child neurologic impairment, as assessed using ASQ3,

**Table 3. Results from testing with the CBCL 1.5–5.**

| | Control (n = 15) | NHIE (n = 14) | *P* value |
|---|---|---|---|
| Internalizing | | | |
| Emotionally reactive | 2.00 (0.76, 2.70) | 1.50 (0.58, 5.56) | 0.43 |
| Anxious/depressive | 1.00 (0.69, 2.76) | 2.00 (1.69, 5.46) | *0.03* |
| Somatic complaints | 1.00 (0.64, 2.01) | 2.00 (0.75, 2.96) | 0.28 |
| Withdrawn | 1.00 (0.64, 2.01) | 0.00 (0.09, 3.34) | 0.19 |
| Externalizing | | | |
| Attention | 2.00 (1.40, 3.39) | 3.00 (1.68, 4.03) | 0.33 |
| Aggressive behavior | 5.00 (4.14, 9.58) | 10.50 (8.27, 16.73) | *0.004* |
| Sleep problems | 2.00 (0.87, 3.38) | 2.50 (1.28, 3.71) | 0.31 |
| Other problems | 6.00 (4.14, 8.78) | 7.00 (4.96, 14.03) | 0.23 |

Median (95% CI). Statistical analysis using the Mann-Whitney test.

CBCL 1.5–5: Child Behavior Checklist 1.5–5.

NHIE: newborn hypoxic-ischemic encephalopathy.

maternal depression as assessed using MADRS, or socio-economic characteristics such as educational level or monthly income (Table 4).

## Discussion

This study alerts about an increased risk of MD, in particular of anxious/depressive nature, as observed in three-to-six-year old children survivors of moderate-to-severe NHIE without CP. This is the first study in which indicators of MD have been studied in children treated with the current standard of care for NHIE, hypothermia, and compared to healthy children with no history of neonatal complications. Increased incidence of MD in NHIE children would link NHIE outcome with that of acute IBI in adults, in which the incidence of MD, in particular depression and anxiety, is dramatically increased [3, 4].

**Table 4. Correlation between developmental assessment and contextual factors and items of mood disturbances in NHIE group children.**

| | PRESS | | | CBCL Anxious/depressed | | | CBCL Aggressive behavior | | |
|---|---|---|---|---|---|---|---|---|---|
| | R | 95% CI | *P* | R | 95% CI | *P* | R | 95% CI | *P* |
| ASQ3 | | | | | | | | | |
| Communication | -0.19 | -0.68, 0.41 | 0.51 | 0.10 | -0.46, 0.61 | 0.72 | -0.05 | -0.54, 0.53 | 0.98 |
| Gross motor | -0.30 | -0.74, 0.31 | 0.31 | 0.10 | -0.46, 0.51 | 0.71 | 0.07 | -0.48, 0.59 | 0.79 |
| Fine motor | -0.26 | -0.72, 0.35 | 0.37 | 0.13 | -0.44, 0.63 | 0.64 | -0.72 | -0.59, 0.49 | 0.80 |
| Problem solving | 0.07 | -0.51, 0.61 | 0.81 | -0.09 | -0.60, 0.47 | 0.75 | -0.17 | -0.65, 0.40 | 0.54 |
| Personal-Social | 0.04 | -0.56, 0.56 | 0.98 | -0.14 | -0.63, 0.43 | 0.61 | -0,52 | -0.82, 0.09 | 0.08 |
| MADRS | 0.32 | -0.29, 0.75 | 0.28 | 0.01 | -0.54, 0.55 | 0.99 | -0.10 | -0.61, 0.46 | 0.72 |
| Education level | -0.56 | -0.89, 0.08 | 0.08 | 0.25 | -0.37, 0.69 | 0.38 | 0.17 | -0.41, 0.65 | 0.56 |
| Monthly Income | -0.47 | -0.82, 0.11 | 0.09 | -0.03 | -0.57, 0.56 | 0.77 | -0.31 | -0.74, 0.43 | 0.29 |

Statistical analysis using Spearman's correlation.

ASQ3: Ages and Stages Questionnaire-3.

CBCL 1.5–5: Child Behavior Checklist 1.5–5.

MADRS: Montgomery-Asberg Depression Rating Scale.

NHIE: newborn hypoxic-ischemic encephalopathy.

PRESS: Preschool Symptom Self-Report.

Cognitive and motor repercussions of NHIE in children have been extensively studied [2, 18]. Cognitive and behavioral impairment, such as difficulties in attention, language and executive functions, are commonly observed in children with NHIE without CP [2]. However, very few studies have included the assessment of socio-emotional disturbances in that population [11, 12]. Assessing MD in pre-school children is difficult because cognitive characteristics and language development at this age lead to a different expression of psychopathology compared to adults [7]. Nevertheless, nowadays there is a consensus regarding the existence and symptomatology of childhood depression [8]. Depressed mood and aggressive behavior are seen as different forms in which children could express depressive symptoms: the Diagnostic and Statistical Manual of Mental Disorders (DSM 5) specifies that, in the case of children, depressive symptoms are usually expressed as irritable mood rather than or in addition to sadness [8].

In this study we used two different scales and different information sources to assess MD and other psychopathologies. The CBCL is one of the most used and well-validated instruments. It is also used to screen MD in children [19], and shows an adequate capacity to detect psychological disorders [20]. In particular CBCL 1.5–5 have proved to be an effective diagnostic tool for children at an age similar to that of the children in our study [21]. It is completed by parents, which enables obtaining comprehensive information about three broad domains: internalizing, externalizing and dysregulation symptoms in which the psychopathology is organized [10]. The other scale, the PRESS, enables obtaining direct information from the children themselves [16], which can facilitate access to internalizing symptoms. Difficulty in accessing these kinds of symptoms from external observation could lead to their underestimation, as compared to externalizing symptoms [16].

Our study is the first to show that survivors of NHIE without CP, even after having been treated with hypothermia, have increased presence of anxiety/depression symptoms, as assessed by two different scales -PRESS and CBCL- as well as aggressive behavior more frequently than healthy children. Previous studies exploring those items detect only increased aggressivity in NHIE survivors with no treatment with hypothermia [11] or only anxiety/ depression symptoms in NHIE survivors treated with hypothermia [12]. Moreover, those studies do not compare NHIE survivors with a normal population but rather compare two subgroups of NHIE survivors, the subgroup with minor or moderate and the other with no neurologic symptoms [11, 12]. Instead of comparing two subpopulations of NHIE children we compared the results of NHIE children with those of healthy children and this likely makes our analysis more powerful. We must be cautious due to the small sample size; but it is worth noting that the number of children from the NHIE group with clear symptoms of depression according to the CBCL -three out of 14- indicates a proportion of depressed children after NHIE similar to the proportion reported for adults after acute IBI [3, 22].

Different conditions could justify the development of MD in children after NHIE. The onset of depressive symptoms in children developing CP after NHIE has been attributed to brain damage and cognitive impairment but not necessarily to the motor disability [9]. In our study, NHIE survivors did not develop CP. In addition, although children from the NHIE group showed worse cognitive and motor performance as observed using the ASQ-3, none showed a frank delay in any item. It is well known that careful examination can unveil subtle worse cognitive and motor performance even in NHIE survivors with no apparent functional repercussion [18]. It was noteworthy that no relationship was revealed in our study between cognitive or motor performance scores and PRESS score or anxious/depressive symptom or aggressive behavior scores for the CBCL. This makes a significant influence of the feeling of living with poor motor and/or cognitive performance on the development of MD unlikely.

Family socio-economic status has a remarkable impact on the onset of motor and cognition development impairment as well as behavioral disturbances after NHIE, in particular when

the developmental impairment is mild [23]. In our study, however, educational background and monthly income were similar in families from NHIE or healthy children, and no relationship was found between those contextual factors and anxiety/depression symptoms in either the CBCL or PRESS score.

The emotional development of children may also be affected by the parents' emotional status. Thus, maternal depression is linked to higher levels of psychopathology and negative affect and behavior in children [24]. In our study MADRS score, which assesses the onset of symptoms of depression in mothers, was similar in mothers of NHIE or of healthy children. Furthermore, no relationship was detected between maternal MADRS score and anxiety/ depression symptoms in either the CBCL or PRESS score.

Altogether, these results indicate that contextual factors such as the appearance of non-severe physical disability, family socio-economic status or maternal depression were not major determinants of the onset of MD in survivors of NHIE. Similarly, external factors such as mental distress due to disability or socio-economic factors are not major determinants of the appearance of MD after acute IBI in adults [5, 25]. Increased evidence supports a neurochemical basis for MD after acute IBI in adults [4, 26]. The most popular hypothesis is that excitotoxicity and inflammation-induced damage of monoaminergic pathways results in reduced synthesis of serotonin and norepinephrine, leading to development of MD [4, 5, 26]. Thus, location of the IBI has a major impact on the risk of subsequent development of MD in adults [3]. Interestingly, in our study all except three HIE newborns had normal MRI scans in the neonatal period; the three abnormal MRI scans consisted of small focal ischemic areas in the parietal cortex, far from the usual location of brain lesions in adults developing MD after IBI [3]. Moreover, all the children in our study had normal MRI scans by the age of two to three years. Immature brain is particularly sensitive to excitotoxicity and inflammation [27]. Hence, hypoxic-ischemic insult might result in monoaminergic pathway damage even in the absence of a visible lesion. However, this remains speculative.

Whereas the comparison of children with NHIE treated with the current standard of care with healthy children is a clear strength of our work, the small sample size of study groups is a limitation. However, the NHIE group size in our work (n = 14) is similar to that of previous studies with NHIE survivors with non-severe neurologic sequelae regardless of cooling (n = 10) [12] or no cooling (n = 17) [11]. Furthermore, NHIE children in our studies attained similar CBCL scores than those of the NHIE subgroup children with minor neurologic sequelae in the study most comparable to ours—the study of children cooled because of NHIE [12]. This similarity adds some robustness to our own results.

The possibility of increased risk of MD as a consequence of NHIE is of paramount importance. In adults, MD after IBI not only increases mortality and impairs quality of life, but clearly has a negative impact on patients' ability to engage in rehabilitation therapies, which jeopardizes their physical and cognitive recovery [4, 26]. Whether or not the onset of MD after NHIE could interfere with effective rehabilitation and healthy development in those children warrants further research.

A limitation of our study could be that researchers were not blind to the experimental group since NHIE children were assessed at the follow-up clinical whereas healthy children attended a special visit. However, the different measures collected in our study were self- completed by parents (ASQ-3, MDRS and CBCL) or children (PRESS). Therefore, since scoring did not need the intervention of the evaluator, it is unlikely that a researcher bias could have affected the results.

## Conclusions

This exploratory work indicates that infants surviving after NHIE treated with hypothermia with non-severe neurologic symptoms and no evidence of brain damage in the MRI, show

increased risk of MD aged three to six when compared with healthy children with a normal neonatal period. These disorders were not related to physical and cognitive developmental scores, family socio-economic status or presence of maternal depression. The relevance of the theoretical impact such disorders could have on children's quality of life and chance of a full recovery justify inclusion of its routine assessment in the follow-up of these children as well as commencing further studies with a larger population.

## Supporting information

**S1 File.**
(PDF)

## Acknowledgments

We thank the children and parents who generously and enthusiastically participated in this study. We are also grateful to Jason Willis-Lee for scientific writing assistance during preparation of the final manuscript.

## Author Contributions

**Conceptualization:** Isabel Cuellar-Flores, Purificación Sierra-García, José Martínez-Orgado.

**Data curation:** María Álvarez-García, Purificación Sierra-García, José Martínez-Orgado.

**Formal analysis:** José Martínez-Orgado.

**Funding acquisition:** José Martínez-Orgado.

**Investigation:** María Álvarez-García, Isabel Cuellar-Flores.

**Methodology:** María Álvarez-García, Isabel Cuellar-Flores, Purificación Sierra-García.

**Supervision:** Isabel Cuellar-Flores, Purificación Sierra-García, José Martínez-Orgado.

**Validation:** Purificación Sierra-García, José Martínez-Orgado.

**Writing – original draft:** José Martínez-Orgado.

**Writing – review & editing:** María Álvarez-García, Isabel Cuellar-Flores, Purificación Sierra-García, José Martínez-Orgado.

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
