## [Decision Letter · Decision Letter 0]

1 Nov 2021

PONE-D-21-13777MOOD DISORDERS IN CHILDREN FOLLOWING NEONATAL HYPOXIC-ISCHEMIC ENCEPHALOPATHYPLOS ONE

Dear Dr. Martínez-Orgado,

Thank you for submitting your manuscript to PLOS ONE. After careful consideration, we feel that it has merit but does not fully meet PLOS ONE’s publication criteria as it currently stands. Therefore, we invite you to submit a revised version of the manuscript that addresses the points raised during the review process.

The reviewers agree that it is a professionally written article that adds new knowledge; therefore, they recommend its publication. However, they do make several suggestions that must be followed.

Additionally, in the statistical analysis, I would like the effect size to be included.

We look forward to receiving your revised manuscript.

Kind regards,

Thalia Fernandez, Ph.D.

Academic Editor

PLOS ONE

Journal Requirements:

Reviewers' comments:

Reviewer's Responses to Questions

**Comments to the Author**

1. Is the manuscript technically sound, and do the data support the conclusions?

Reviewer #1: Yes

Reviewer #2: Yes

2. Has the statistical analysis been performed appropriately and rigorously? 

Reviewer #1: Yes

Reviewer #2: No

3. Have the authors made all data underlying the findings in their manuscript fully available?

Reviewer #1: Yes

Reviewer #2: Yes

4. Is the manuscript presented in an intelligible fashion and written in standard English?

Reviewer #1: Yes

Reviewer #2: Yes

5. Review Comments to the Author

Reviewer #1: This manuscript is professionally written. It describes the possible presence of Mood Disorders (MD) in a group of 14 children between 3- and six-year-old that at birth present Hypoxic Ischemic Encepalopathy (HIE) from moderate to severe (the scale values of Sarnat and Sarnat scores are not described) that were treated with hypothermia and without cerebral palsy and 15 healthy children without perinatal complications matched by gestational age, socioeconomic status, parental educational level, or monthly income. Infants had 35 weeks of gestational age or more. I will recommend the authors to include more clearly the number of preterm and term infants in the study, since the mean age in both groups was 38 weeks.

The study used well known questionnaires to the parents, the Ages and Stages Questionnaire 3 (ASQ-3), the Child Behavior Checklist (CBCL) for children aged one and a half to five years, and the Montgomery-Asberg Depression Rating Scale (MADRS). Children completed the Preschool Symptom Self-Report (PRESS), a 143 pictorial instrument that assesses, with self-informed responses from identification with 144 images, the presence of depression symptoms in preschool children.

Unfortunately, in the abstract only the acronyms of the tests are used. My recommendation is to use the complete title of the tests also in the abstract. This is mandatory.

Magnetic Resonance Images (MRI) were obtained from the children early after birth and at the age of two to three years. Authors only described the results of the images, but not include any figure. I think that the presentation of the images of the child with a lesion at birth and that was normal after 3 years will be welcome. The authors only describe MRI studies of the HIE group. To have a perfect mismatch with the HIE children, studies of the healthy children should be also included. It has been shown by MRI (Rutherford et al., 2010) that hypothermia treatment of infants with HIE decreases brain injury tissues, therefore should be interesting to know the MRI results of the control group. In case of preterm infants frequent MRI abnormalities have been described (Volpe, J in 2011, 2019; Woodward et al. 2012; and Harmony, 2021).

Conclusion: I recommend the authors to follow my suggestions.

Reviewer #2: Dear Editor and the authors,

Thank you for the opportunity to review the report about mood disorders in children with hypoxic-ischemic encephalopathy (HIE). The report is well written and worth publication, even it is a descriptive study because of the paucity of knowledge about emotional developments of infants after neonatal HIE. It adds new knowledge and is worth publication. However, there are some limitations of study, which have not been discussed. I will specify it below.

First, I will address the statistical analysis and presentation of the results. The authors have chosen the right method for the analyses (Mann-Whitney U test) because of skewed distribution of data. Therefore, results should be presented by medians instead of means.

The study was well planned, with two different sources of data regarding mood disorders (CBCL and PRESS) and a control group. I will also commend the authors for including the most important environmental factors (education level, income and parental depressive symptoms). Unfortunately, the assessors were not blinded. I fully understand that it is difficult to blind the assessors in a clinical setting, but I really miss discussion how not-blinded evaluation of the outcome affects the results.

Line 69-71. The authors state that emotional consequences of disability of CP can mask the effects of NHIE by itself. I totally agree. Later, when the authors cite the same study in the discussion, Line 281, they interpret results differently than the authors of the cited study. Rackauskaite et al. wrote “the high prevalence of psychopathology in children with CP may be due to brain impairment or cognitive disability and not to the motor disability itself”. Please, correct it.

Line 160-164 presents data from the Figure 1. There is no need for repetition.

Line 182/ Table 1. There is a difference of 6 months between the mean age of control and NHIE. Emotional development is quite fast between the age of 4½ and 4 years. Please, discuss if that can influence your results, even the ages are not statistically significant.

Line 238 is difficult to read. Please, rephrase.

Line 270-275 repeats information from the introduction.

Line 332-334. Handicap is a strong word in the context of HNIE. I suggest to reword “could be a handicap” by using a verb like “could disturb” or “interfere with”.

Line 341-342. Your data does not support the statement, that MD were not related to the severity of handicap. All children (HNIE and controls) had a normal motor and cognitive function, thereby no handicap.

I hope my suggestions will help the authors to improve their report.

6. PLOS authors have the option to publish the peer review history of their article (what does this mean?). If published, this will include your full peer review and any attached files.

Reviewer #1: **Yes: **Thalía Harmony

Reviewer #2: No

---

## [Author Response · Author response to Decision Letter 0]

18 Nov 2021

PONE-D-21-13777

Answers to Reviewer #1: 

We wish to thank the Reviewer because his/her comments have greatly improved the quality of the manuscript. 

Comments:

- I will recommend the authors to include more clearly the number of preterm and term infants in the study, since the mean age in both groups was 38 weeks.

o This has been included in the revised manuscript (lines 194-195). Since the original statement was confusing -all babies were of more than 35 weeks of gestational age-, it has been rewritten (line 107). 

- My recommendation is to use the complete title of the tests also in the abstract.

o Complete title of tests have been included in the Abstract

- Magnetic Resonance Images (MRI) were obtained from the children early after birth and at the age of two to three years. Authors only described the results of the images, but not include any figure. I think that the presentation of the images of the child with a lesion at birth and that was normal after 3 years will be welcome. The authors only describe MRI studies of the HIE group. To have a perfect mismatch with the HIE children, studies of the healthy children should be also included. It has been shown by MRI (Rutherford et al., 2010) that hypothermia treatment of infants with HIE decreases brain injury tissues, therefore should be interesting to know the MRI results of the control group. In case of preterm infants frequent MRI abnormalities have been described (Volpe, J in 2011, 2019; Woodward et al. 2012; and Harmony, 2021).

o Representative examples of abnormal MRI in NHIE newborns with the correspondent normal scan when aged 2-3 years have been included in the revised version of the manuscript (Figure 2) (lines 182-189).

o The aim of the present work was focused on the study of Mood Disorders in the hypoxic-ischemic babies. The aim of presenting the MRI studies was to explore the possibility that mood disorders were related to the presence of damage in specific brain regions, which was not the case. Our work was not designed to explore the effects of hypothermia on hypoxic-ischemic brain damage as assessed by MRI, which has been extensively studied already, as the Reviewer stated. That is why the Ethical Committee did not approve to perform MRTI studies on healthy babies and children, who should have been exposed to sedation for the MRI study to be performed.

o MRI abnormalities are present mostly in very preterm infants (under 32 weeks of gestational age), as described in the bibliography cited by the Reviewer. Thus, the American Association of Neurology recommends since 2004 that routine MRI studies were performed only in very preterm infants or in those newborns at high neurological risk (Ment L et al, 2004). In our case, the only baby born at 36 weeks of gestational age in the NHIE group showed no abnormalities at the MRI studies. 

 

PONE-D-21-13777

Answers to Reviewer #2: 

We wish to thank the Reviewer because his/her comments have greatly improved the quality of the manuscript. 

Comments:

- Results should be presented by medians instead of means

o All results have been presented as medians in tables and text in the revised manuscript

- I fully understand that it is difficult to blind the assessors in a clinical setting, but I really miss discussion how not-blinded evaluation of the outcome affects the results.

o The reviewer points out a very relevant aspect of clinical research. Having into account that NHIE children were assessed at the follow-up clinical whereas healthy children attended a special visit, blinding was impossible. However, the different measures collected in our study were either completed by parents or children (therefore without the intervention of the evaluator) or are highly structured and, again, do not require the interpretation of the researcher. Thus, Ages and Stages Questionnaire 3 (ASQ-3) and Child Behavior Checklist (CBCL) are questionnaires answered by parents, whereas the Preschool Symptom Self-Report (PRESS) scores as “present” or “absent” the answers of the child to a series of drawings. The Montgomery-Asberg Depression Rating Scale (MADRS) is a self-reported questionnaire. Therefore, researcher’s knowledge of about the experimental group is unlikely to interfere with the score since this was independent from the researcher’s opinion. This, lack of blinding is usual in this kind of studies (please see refs. 11 and 12 of the manuscript). This has been included in the Discussion (lines 441-447).

- Line 69-71. The authors state that emotional consequences of disability of CP can mask the effects of NHIE by itself. I totally agree. Later, when the authors cite the same study in the discussion, Line 287, they interpret results differently than the authors of the cited study. Rackauskaite et al. wrote “the high prevalence of psychopathology in children with CP may be due to brain impairment or cognitive disability and not to the motor disability itself”. Please, correct it.

o This has been corrected in the manuscript (lines 386-387).

- Line 160-164 presents data from the Figure 1. There is no need for repetition.

o This has been so corrected in the revised version of the manuscript (line 170)

- Line 182/ Table 1. There is a difference of 6 months between the mean age of control and NHIE. Emotional development is quite fast between the age of 4½ and 4 years. Please, discuss if that can influence your results, even the ages are not statistically significant.

o Besides the lack of statistical difference respecting the age at evaluation, age rank in both populations was very similar. In addition, CBCL 1.5-5 applies the same questionnaire for children between 1.5 and 5 years of age, assuming that responses between those boundaries are not affected by age (please see ref. 15 of the manuscript). In the case of ASQ-3 and PRESS, these are standardized tests, i.e., normalized for age. Therefore, the influence of the age difference in the results of our study is unlikely. In fact, even broader age ranks can be found in studies similar to ours (please see refs 11 and 12 of the manuscript). 

- Line 238 is difficult to read. Please, rephrase.

o The line has been rephrased in the revised version of the manuscript (lines 338-339)

- Line 270-275 repeats information from the introduction.

o Repeated information has been removed from the Introduction in the revised version of the manuscript (line 87).

- Line 332-334. Handicap is a strong word in the context of HNIE. I suggest to reword “could be a handicap” by using a verb like “could disturb” or “interfere with”.

o This has been so reworded in the revised version of the manuscript (line 439)

- Line 341-342. Your data does not support the statement, that MD were not related to the severity of handicap. All children (HNIE and controls) had a normal motor and cognitive function, thereby no handicap.

o A more appropriate statement has been included in the revised version of the manuscript (lines 454-455)

---

## [Editor Report · Decision Letter 1]

12 Jan 2022

MOOD DISORDERS IN CHILDREN FOLLOWING NEONATAL HYPOXIC-ISCHEMIC ENCEPHALOPATHY

PONE-D-21-13777R1

Dear Dr. Martinez-Orgado,

We’re pleased to inform you that your manuscript has been judged scientifically suitable for publication and will be formally accepted for publication once it meets all outstanding technical requirements.

Kind regards,

Thalia Fernandez, Ph.D.

Academic Editor

PLOS ONE
---

## [Editor Report · Acceptance letter]

20 Jan 2022

PONE-D-21-13777R1 

Mood Disorders in Children following Neonatal Hypoxic-Ischemic Encephalopathy 

Dear Dr. Martínez-Orgado:

I'm pleased to inform you that your manuscript has been deemed suitable for publication in PLOS ONE. Congratulations! Your manuscript is now with our production department. 

Kind regards, 

on behalf of

Dr. Thalia Fernandez 

Academic Editor

PLOS ONE